# A Novel Functional Emulsifier Prepared with Modified Cassava Amylose with Octenyl Succinic Anhydride and Quercetin: Preparation and Application in the Pickering Emulsion

**DOI:** 10.3390/molecules26226884

**Published:** 2021-11-15

**Authors:** Hailing Zhang, Haiming Chen, Shan Jiang, Xiaoning Kang

**Affiliations:** 1College of Life Sciences, Yantai University, 30 Qingquan Road, Yantai 264005, China; hlzhang@ytu.edu.cn; 2Maritime Academy, Hainan Vocational University of Science and Technology, 18 Qingshan Road, Haikou 571126, China; 3Key Laboratory of Food Nutrition and Functional Food of Hainan Province of China, Hainan University, 58 People Road, Haikou 570228, China; 15848911030@163.com; 4Haikou Key Laboratory of Areca Processing and Research, Hainan Academy of Agricultural Sciences, 14 Xingdan Road, Haikou 571100, China

**Keywords:** emulsifier, antioxidant, OSA starch, emulsion, quercetin

## Abstract

An emulsifier with a targeted antioxidant effect was prepared using the inclusion complexes of octenyl succinic anhydride (OSA)-modified cassava amylose (CA) and quercetin (Q). The designed emulsifier, a carbohydrate polymer-flavonoid complex, exhibited both amphiphilic and antioxidant properties. To investigate the physical and oxidation stabilities of the prepared emulsion, three types of emulsions were prepared: primary emulsions stabilized by enzyme-modified starch, secondary emulsions stabilized by OSA-CA, and tertiary emulsions stabilized by Q-encapsulated complexes (OSA-CA/Q). The structural characteristics of CA, OSA-CA, and OSA-CA/Q were investigated by scanning electron microscopy, Fourier transform infrared spectrometry, and small-angle X-ray scattering analysis. The stabilities of the emulsions were evaluated based on their particle size distribution, zeta potential, creaming stability, and peroxide value. The results showed that the secondary and tertiary emulsions exhibited a relatively narrower particle size distribution than the primary emulsions, but the particle size distribution of the tertiary emulsions was the narrowest (10.42 μm). Moreover, the secondary and tertiary emulsions had lower delamination indices than the primary emulsions after 7 days of storage. The results obtained from the antioxidant experiments indicated that OSA-CA/Q exhibited good oxidation stability for application in emulsion systems.

## 1. Introduction

Emulsions are thermodynamically unstable systems and are widely used in many fields such as cosmetics and food industries [1,2]. Therefore, in order to prepare an emulsion that is relatively stable for a certain period of time (shelf life or service life), emulsifiers must be added. In addition, antioxidants are also added to certain types of emulsions to avoid the health risks and undesirable flavor associated with lipid oxidation. Generally, emulsion systems can be divided into three parts: the interior of the droplet, the continuous phase, and the interfacial membrane [3,4]. Many studies have reported that the interface is the main site of oxidation reactions in emulsions [5,6,7,8,9]. Therefore, the effectiveness of chain-breaking antioxidants to retard lipid oxidation in oil-in-water (O/W) emulsions increases as the polarity of the emulsions decreases or their surface activity increases [10,11]. Previous studies have suggested that amphiphilic antioxidants are the most effective for retarding or preventing lipid oxidation.

Emulsifiers, which are amphiphilic molecules, facilitate the formation of emulsions and improve emulsion stability by decreasing the oil–water interfacial tension and increasing the steric hindrance and/or electrostatic repulsion between the droplets [12,13]. The most commonly used emulsifiers in the food industry are proteins, phospholipids, and polysaccharides. For instance, the emulsifying properties of cyclodextrins have been utilized to improve the quality of processed foods such as mayonnaise, condiments, and whipped cream [14]. As an emulsifier, xanthan gum enhances the stability of soybean oil emulsions with egg yolks [15]. Pickering emulsions can be obtained when solid particles, which absorb onto the oil–water interface, are used instead of surfactants [16,17]. The strong adsorption of particles at the interface prevents the direct physical contact of the oil–water interface of oil droplets in Pickering emulsions [18], and these emulsions show irreversible adsorption compared with surfactants [19,20]. The improved stability of Pickering emulsions is attributed to these properties.

Previous studies have reported the following three types of solid particles as ideal for Pickering emulsion preparation: inorganic salts (e.g., SiO_2_ [21] and Fe_3_O_4_ [22]), nanocomposites (e.g., nanosheets and CNT [23]), and biodegradable materials (e.g., starch [24] and cellulose [25]). There are two approaches to obtain more ideal solid particles: the development of new solid particles and the modification of existing solid particles [26,27]. Among the methods for the second approach, the hydrophobic modification of starch particles by using octenyl succinic anhydride (OSA) [28] has been widely applied in the food industry as it promotes the preservation of the biodegradability of the starch [29].

It is known that native starches consist of two main components: amylose having an essentially linear molecular structure with few branches, and amylopectin having a highly branched structure. Amylose structures comprise single, left-handed helices with a hydrophilic exterior and a hydrophobic helix cavity, which enables its interactions with hydrophobic compounds to form amylose-inclusion complexes [30,31]. However, amylose as a wall material has not been widely applied. In general, there are three main synthesis methods for amylose-inclusion complexes: the direct use of starch or amylose, the mixing of amylose and the desired ligand, and the utilization of ligand-added amylose chains. The amylose-inclusion complexes prepared using the last two methods are purer than those using the first method [32].

Hence, in our study, the pullulanase-treated cassava starch was used as the wall material. Quercetin (3,3′,4′,5′-7-pentahydroxy flavone), one of the most prominent dietary antioxidants, which has important biological activities [33], low aqueous solubility [34], and low bioavailability [35,36,37,38], acted as the ligand. Therefore, a new emulsifier with a targeted antioxidant effect was prepared by mixing inclusion complexes of OSA-modified starch with quercetin. The structures of the inclusion complexes were characterized by small-angle X-ray scattering (SAXS) analysis and Fourier transform infrared (FT-IR) spectroscopy. Furthermore, the applicability of the prepared emulsifier as a Pickering emulsion was verified. The properties of the prepared emulsifier were characterized by measuring various physicochemical properties that are related to the physical and oxidation stability of the emulsions, such as zeta potential, particle size, creaming index, and peroxide value. This study provides a multifunctional stabilizer, which can reduce the quantity of antioxidants or even replace them in O/W emulsions.

## 2. Results and Discussion

### 2.1. SEM Analysis

Figure 1 shows that native cassava starch (Figure 1A-1) was an ellipsoid of different sizes, some of which were irregular in shape. Their surfaces were smooth and flat, and the particles were relatively independent and evenly distributed (Figure 1A-2).

After enzyme modification, the starch had a rough surface with cracks, irregular distribution and shape, and the spherical particles were broken. The enzymatic modified starch was irregular (Figure 1B-1), which was due to the gelatinization of raw starch granules at high temperatures. Enzymatic hydrolysis of native starch magnified 20,000 times (Figure 1B-2) showed that the starch surface, which had been smooth, became a more prominent chain structure. This could be explained by the fact that most of the branches in the native starch have been hydrolyzed by pullulanase. The CA surface (Figure 1C-1) treated by OSA was uneven, and the irregularly dispersed unit volume was larger than that of the native cassava starch, indicating that the esterification reaction between OSA and CA increased the unit mass of starch. This is because the esterification with OSA occurred on the CA surface, so that the surface became rough and irregular (Figure 1C-2). The OSA-CA/Q (Figure 1D-1) has a lower dispersion than OSA-CA and has a rougher surface (Figure 1D-2) and a more complex structure. These results indicate that quercetin particles were encapsulated in OSA-CA and most of the quercetin was attached to the surface of the starch.

### 2.2. SAXS

As seen from Figure 2a, the SAXS peak at *q* = 0.58 nm^−1^ indicated that most native starches are two-dimensional, semi-crystalline structures, consisting of alternating non-crystalline and crystalline regions. According to the relation *d = 2π/q* and scattering vector *q*, the semi-crystalline layer thickness *d_CS_* of the cassava starch was calculated to be 10.83 nm [39,40]. Compared with curve a, the curve of CA (Figure 2b) was smoother and its peak at *q* = 0.58 nm^−1^ was not observed, suggesting that the semi-crystalline structure of the cassava starch was destroyed by pullulanase and was unobservable. Moreover, it is suggested that the effect of pullulanase debranching on gelatinized cassava starch was good and that amylose had a higher purity. As shown in Figure 2c, the OSA-CA curve was almost the same as that of enzyme-modified starch, except for a slight decrease in the scattering intensity, which indicated that esterification with OSA had little effect on the degree of crystallization of the enzyme-modified starch. However, after the formation of the inclusion complexes of OSA-CA with quercetin, there was a new SAXS peak at *q* = 0.64 nm^−1^ (Figure 2d). This indicated the formation of a new, semi-crystalline structure, which is in line with the structural characteristics of quercetin as a yellow acicular crystal and further verified the inclusion of OSA-CA with quercetin. The calculation reveals that the value of the semi-crystalline layer thickness was *d*_OSA-CA/Q_ = 9.82 nm, which is slightly lower than that of cassava starch. This could be because the formation of the two semi-crystalline structures is different [41].

### 2.3. FT-IR Analysis

As seen from Figure 3b, the extremely broad band at 3417 cm^−1^ and the peak at 2931 cm^−1^ were attributed to polysaccharide-typical O–H association and C–H stretching, respectively. The peak at 1731 cm^−1^ was attributed to the O–H-bending vibration. In the fingerprint region of the starch spectrum, there were three discernible peaks at 1017, 1082, and 1158 cm^−1^ and peaks around 576 cm^−1^, which were attributed to C–O stretching and basic C–C skeleton vibration. As shown in Figure 3c, the peak at 1731 cm^−1^ that originated from the O–H-bending vibration was not observed in the esterified starch, and a new peak occurred near the original, indicating the esterification of enzyme-modified starch with OSA. Thus, the peak at approximately 1572 cm^−1^ corresponds to the asymmetric stretch of vibration of RCOO−, indicating that OSA groups were successfully bonded to CA.

In the spectrum of quercetin (Figure 3f), the broad band at 3408 cm^−1^ corresponds to O–H stretching vibration, suggesting that there were many hydroxyl groups in quercetin, while the peak at 1665 cm^−1^ corresponds to O–H bending. Strong peaks at approximately 1611 and 1522 cm^−1^, as well as a mid-strong peak at 1561 cm^−1^, were characteristic of C=C stretching vibrations in the aromatic ring. The absorption peaks in the fingerprint region between 1262 and 603 cm^−1^ were relatively dense, with the characteristics of specific fingerprints.

Most of the adsorption peaks of OSA-CA (Figure 3c), the mixture (Figure 3d), and OSA-CA/Q (Figure 3e) were similar to each other; however, the intensity of the mixture (Figure 3d) was weaker than that of OSA-CA (Figure 3c) and OSA-CA/Q (Figure 3e). In addition, compared with Figure 3c,d, three characteristic peaks of OSA-CA/Q (Figure 3e) moved to a high band, i.e., 1662→1731 cm^−1^, 1565→1645 cm^−1^, 1523→1568 cm^−1^, which suggests the formation of inclusion complexes of OSA-CA with quercetin.

### 2.4. Droplet Size and Morphological Observations

The droplet size distributions are shown in Figure 4e. The particle size distribution of the three emulsions (PE, SE, and TE) showed a single peak and was mainly distributed around 10 μm. However, both SE and TE exhibited a narrower particle size distribution than PE, which indicated that SE and TE were more uniform. In addition, both SE and TE had more small droplets due to the introduction of OSA groups. It can be seen from the optical microscopy images (Figure 4a–d) that both SE and TE have a denser and more uniform droplet distribution. In our case, the possible cause of this phenomenon could be the intertwining of the esterified starch at the oil–water interface and the aqueous phase, increasing the density between the droplets.

### 2.5. Fluorescence Microscope

The emulsion, which was prepared by a high-speed shear emulsification using OSA-CA/Q as a stabilizer, was observed under a fluorescence microscope. Figure 4d shows that OSA-CA/Q adsorbed well onto the surface of the oil droplets, indicating the formation of a Pickering emulsion. This observation also suggests that quercetin encapsulation complexes have the potential to be used as emulsifiers for stabilizing edible oil-in-water emulsions, without the presence of other co-surfactants.

### 2.6. Interface Charges

Zeta potential, a measure of mutual exclusion or attraction between emulsion particles, is an important indicator of the electrostatic stability of an emulsion system. As shown in Figure 5, the zeta potential of enzyme-modified starch was −4.2 ± 0.6 mV. Compared with enzyme-modified starch, the zeta potential of OSA-CA was relatively high (−11.5 ± 0.4 mV) after the hydrophobic modification, indicating an increase in the charge density of the OSA-CA surface and in the electrostatic stability of the emulsion [42]. After the inclusion of quercetin, the zeta potential of OSA-CA/Q decreased to −17.8 mV, which may have been due to the relatively high concentration of this flavonoid within particles. However, as reported by Pool et al., particle charge is not strongly affected by the presence of quercetin [43].

### 2.7. Creaming Stability Measurement

From the results (Figure 6) obtained for creaming, it was concluded that the emulsions with quercetin encapsulation complexes achieved the lowest values of the creaming index during the fifth day of storage. According to Xu et al. [44], the stabilization of emulsions is mainly contributed by the starch structure. The starch is esterified with OSA to introduce a hydrophobic alkenyl group and a hydrophilic carboxylic acid group, both of which have improved thickening and emulsifying properties. The SE and TE had lower delamination indices than the PE at 7 days of storage, indicating that the emulsification of the esterified starch was increased. It is well known that starch is a hydrophilic molecule, and the introduction of OSA groups can increase the hydrophobicity of starch, thereby obtaining a better hydrophilic–hydrophobic balance. This was essential for highly stable emulsions [45]. In addition, the higher continuous-phase viscosity can delay the movement of oil droplets, thereby inhibiting the creaming process [46]. Notably, the starch that was esterified but not embedded with quercetin was the least emulsifiable at 12 and 15 days of storage. This is probably due to the oxidation of the droplet surface of the emulsion to the starch granules. The adhesion is lowered, resulting in a decrease in the performance of the emulsifier.

### 2.8. Oxidative Stability of Emulsions

The peroxide values (POVs) of the three kinds of emulsion are shown in Table 1. As expected, the POV values of the emulsions containing OSA-CA/Q were the lowest, which was mainly due to the antioxidant property of quercetin. Due to its structural features, i.e., a 3′, 4′-orthodihydroxy configuration in the B ring and 3-OH and 5-OH in the C ring groups, quercetin showed an obvious antioxidant activity based on significantly (*p* < 0.05) lower POV values [47]. Moreover, according to the paradox theory, quercetin, an oil-soluble antioxidant, is more effective for oxidative resistance in O/W emulsions [48,49]. The POVs of the emulsions stabilized by enzyme-modified starch and OSA-CA were approximate, but the values of the OSA-CA-stabilized emulsions were relatively lower. This may have been due to the unsaturated bonds in OSA, which also have a certain antioxidant effect. In the emulsions stabilized by enzyme-modified starch and OSA-CA, the POV values reached the peak values of 14.0 and 10.0 at 7 days of storage, after which the values decreased to 12.4 and 8.9 at 15 days of storage. These decreases in POVs were presumed to be due to the decomposition of primary oxidation products, i.e., lipid hydroperoxides decomposed into secondary oxidation products, including aldehydes, ketones, and acids [50]. Therefore, OSA-CA/Q has the potential to be used as an antioxidant in O/W emulsions.

## 3. Materials and Methods

### 3.1. Materials

Commercial food-grade cassava starch was obtained from the Taihua Group (QR-CS12, Foshan, China). Pullulanase (E.C. 3.2.1.1, activity 1000 units/g) and glucoamylase from *Aspergillus niger* was supplied by Genenco Bioengineering Co., Ltd. (Wuxi, China). Quercetin was provided by Shandong Xiya Chemical Industry Co. Ltd. (Shandong, China). Sodium acetate was purchased from Boyan Biotechnology Co., Ltd. (Zhengzhou, China). OSA with a minimum purity of 95% was provided by Sigma-Aldrich (St. Louis, MO, U.S.A.). Camellia seed oil was obtained from Yihai Kerry (Hefei, China). Fluorescent dyes (Nile Red and Nile Blue A) were obtained from Source Leaf Biotechnology Co. Ltd. (Shanghai, China). All other chemicals were supplied by Guangzhou Chemical Reagent Factory (Guangzhou, China) and were analytical reagent grade commercial products.

### 3.2. Preparation of Cassava Amylose (CA) Starch

Cassava starch (32 g, dry basis) was mixed with 400 mL of acetic acid buffer (pH 5.5) in a conical flask (500 mL) and gelatinized in a boiling water bath for 30 min. After gelatinization, the mixture was cooled to 60 °C and the debranched reaction was initiated by adding the pullulanase (10 ASPU/g starch). To completely debranch the starch, the solution was incubated at 55 °C with a magnetic stirrer for 24 h. The sample was placed in a boiling water bath for 20 min to deactivate the enzyme and then cooled to room temperature. After drying and crushing with a 100-mesh sieve, the CA powder was prepared.

### 3.3. Preparation of OSA-Modified CA (OSA-CA)

A 30% solution of CA was prepared by dissolving a certain mass of amylose in distilled water while stirring at room temperature. The pH of the solution was adjusted to >9.0 using NaOH (5 wt%). A total of 4–5 drops of octenyl succinic anhydride (OSA) were added to the solution. At the same time, changes in the pH were observed and the pH was maintained at >9.0 while the mixture was stirred at indoor temperature by using a thermostatic magnetic stirrer (DF-101S, Shanghai Yushen Instrument Co. Ltd., Shanghai, China). After some time, OSA was added again slowly and stirred well in an alkaline environment. The total amount of OSA added was calculated using the following Equation (1):(1)$$m_{2}=3\%\cdot m_{1}$$
where *m*_1_ is the mass of amylose (g) and *m*_2_ is the mass of OSA (g).

After the CA was hydrophobically modified with OSA within 2 h, the pH of the mixture was adjusted to 6–7 using 3 wt% HCl. Then, OSA-CA was obtained using a freeze dryer (LABCONCO, Beijing Zhao Shengxing Instrument Equipment Co. Ltd., Beijing, China).

### 3.4. Preparation of OSA-CA/Q

Different contents (1, 3, 5, 7, and 9 wt% of dry weight of amylose) of quercetin and a certain mass of amylose were dispersed in the corresponding volume (the ratio of quercetin mass to absolute ethyl alcohol = 4.07 mg:1 mL) of absolute ethyl alcohol with continuous stirring. The pH of the system was adjusted to 5.0, 6.0, 7.0, 8.0, and 9.0 using 3 wt% HCl or 5 wt% NaOH solution. A thermostatic magnetic stirrer (DF-101S, Shanghai Yushen Instrument Co. Ltd., Shanghai, China) was used to maintain the temperature of the reaction environment at 30, 40, 50, 55, and 60 °C for 1, 2, 3, 4, 5, and 6 h, respectively, and the inclusion complex products (OSA-CA/Q) were obtained.

### 3.5. Preparation of Emulsions

The emulsions were prepared with 5% Camellia seed oil and different surfactants (CA, OSA-CA, OSA-CA/Q) with a concentration of 8% relative to the water mass. To avoid microbiological contamination, sodium azide (0.01%) was added to yield a final concentration of 0.01 wt%. Then, the resultant mixtures were sheared using an Ultra Turrax T18 homogenizer (IKA Labortechnik, Staufen, Germany) at 10,000 rpm for 5 min at room temperature to prepare the emulsions.

### 3.6. Scanning Electron Microscopy (SEM)

The homogeneity and morphology of the native cassava starch, CA, OSA-CA, and OSA-CA/Q samples were observed by using a scanning electron microscope (SIGMA300, Carl Zeiss Co., Ltd., Jena,, Germany). The samples were mounted onto bronze stubs and coated with a gold layer. The images were collected at an accelerating voltage of 5.0 kV. Micrographs were recorded at 2000× and 20,000× magnification to obtain clear images.

### 3.7. Small Angle X-ray Scattering Analysis

SAXS measurements were performed on a SAXSess camera (Anton-Paar, Graz, Austria) connected to an X-ray generator (Philips, PW3830, Eindhoven, The Netherlands) operating at 40 kV and 50 mA with a sealed-tube copper anode, using a Cu Kα radiation wavelength of 1.54 Å as the X-ray source. Starch slurries were prepared by adding samples and deionized water (4:6, wt%). Starch slurries were made to stand for 24 h and then transferred into glass capillary tubes. The measurements were carried out under vacuum to minimize air scatter [39,40,51,52,53]. The background scattering and smeared intensity were eliminated using the SAXSquant 3.0 software for effective data analysis. According to the Bragg’s law, in the SAXS application, the average repeat distance of the amorphous and crystalline lamellae of each sample and the relationship between *q* and *θ* were calculated using Equation (2) and (3):(2)$$d=\frac{2\pi }{q},$$
(3)$$q=\frac{4\pi }{\lambda }sin\theta ,\;$$
where *d* is the thickness of the semi-crystalline layer, *q* is the scattering vector, and *2θ* is the scattering angle.

### 3.8. FT-IR Spectroscopy

The FT-IR spectra were acquired using an IR-960 spectrophotometer (Tianjin Ruian Technology Co. Ltd., Tianjin, China.). Starch powder was blended with anhydrous KBr powder. The spectra were obtained with a resolution of 4 cm^−1^ in the wavenumber range of 400–4000 cm^−1^ [54,55].

### 3.9. Morphological Observations

The microstructures of the prepared emulsions were observed using an Olympus CX31 optical microscope (Olympus Corporation, Tokyo, Japan). A drop of the emulsion was placed on a glass microscope slide and covered with a cover slip and observed under a microscope at 100× magnification.

To investigate the adsorption of OSA-CA/Q on the emulsion interface, a fluorescence microscope (XD-RFL, Ningbo Shunyu Instrument Co. Ltd., Ningbo, China) was employed.

### 3.10. Zeta Potential Measurements

The interface charge of the emulsion droplets was determined by zeta potential measurement, which was carried out using a Zetasizer Nano ZS90 (Malvern Instruments Let., Malvern, UK) at 25 °C. Emulsions were diluted 100-fold with deionized water to avoid multiple scattering effects. The diluted emulsions were mixed thoroughly and then injected into the instrument. A parallel-plate electrode was inserted, and the cuvette was placed in a temperature-controlled holder. The zeta potential of each sample was calculated from the average of five measurements of the diluted emulsion. The results are reported as the mean and standard deviation.

### 3.11. Droplet Size Distribution

The droplet size of emulsions was measured using a laser scattering particle size analyzer (WJL-602, Shanghai INESA Scientific Instrument Co., Ltd., Shanghai, China). The newly prepared emulsion (1 mL) was added to the sample cell. The refractive indices of Camellia seed oil and water are 1.47 and 1.33, respectively.

### 3.12. Creaming Stability Measurement

The emulsion samples were created at room temperature for a period of 15 days. The volume of the transparent serum layer formed at the bottom of the cylinder could be noticed visually and expressed by *h_S_*. The total volume of the emulsion sample was *h_T_*. The extent of creaming was characterized by the creaming index (%), Equation (4):(4)$$CI=\frac{h_{S}}{h_{T}}\times 100\%.$$

### 3.13. Measurement of Peroxide Value (POV)

POVs of the emulsions were measured after storage for 0, 1, 3, 5, 7, and 15 days at room temperature in the dark, according to the AOCS standard procedure Cd-8-53 [56,57]. Briefly, approximately 2 mL of the sample was mixed with 30 mL acetic acid-chloroform (3:2, *v*/*v*) and 1 mL saturated KI. After the solution was mixed, water and starch indicators were added to the solution. Then, it was titrated with 0.002 N standard sodium thiosulfate (Na_2_S_2_O_3_) and shaken vigorously until the blue color disappeared. Water was used instead of the sample in the control experiment, and the POV was calculated using Equation (5):(5)$$\mathrm{POV}\left(meq\;O_{2}\;kg\;oil^{-1}\right)=\frac{\left(V_{S}-V_{B}\right)\times N\times 1000}{W},$$
where *V_S_* is the volume of the standard sodium thiosulfate solution taken up by the sample (mL), *V_B_* is the volume of standard sodium thiosulfate solution taken up by the control sample (mL), *N* is the concentration of standard sodium thiosulfate solution (mol/L), and *W* is the mass of the sample (g). All POVs were determined in triplicate and mean values are given.

## 4. Conclusions

In this study, OSA-modified starch was obtained by hydrophobic modification of cassava starch with OSA. OSA-CA/Q was prepared by the inclusion reaction of OSA-CA with quercetin. The chosen optimum conditions, which were used for further analyses, were as follows: reaction temperature 55.29 °C, pH of reaction system 7.13, and additive amount of quercetin 5.29% (wt%). Under theoretical conditions, the predicted quercetin content was 24.654 mg/g. From an operational perspective, the reaction temperature was 55 °C, the pH of the reaction system was 7.1, and the amount of quercetin was the same. Under these conditions, the quercetin content was 24.36 mg/g, which is close to the theoretical value. Further, SEM, SAXS, and FT-IR analyses confirmed the formation of inclusion complexes.

PE-, SE-, and TE-stabilized emulsions were prepared to investigate the physical and oxidation stabilities of the emulsions. The results obtained from droplet size and microscopic observations showed that OSA-CA and OSA-CA/Q are suitable emulsifiers for Pickering emulsifiers because their amphiphilic characteristics increase the emulsifying properties of emulsifiers. Furthermore, the zeta potential and creaming stability of the TE showed that OSA-CA/Q provided good stability for emulsion systems. In addition, the quercetin embedded in the OSA-CA is transported to the oil–water interface, which can enhance the oxidation stability of the emulsion according to the results of antioxidant experiments.

In summary, OSA-CA/Q, i.e., the inclusion complexes of OSA-CA with quercetin, provided physical and oxidation stability for emulsion systems. The effect of molecular weight on the SE and TE will be addressed in future studies.

## Figures and Tables

**Figure 1 molecules-26-06884-f001:**
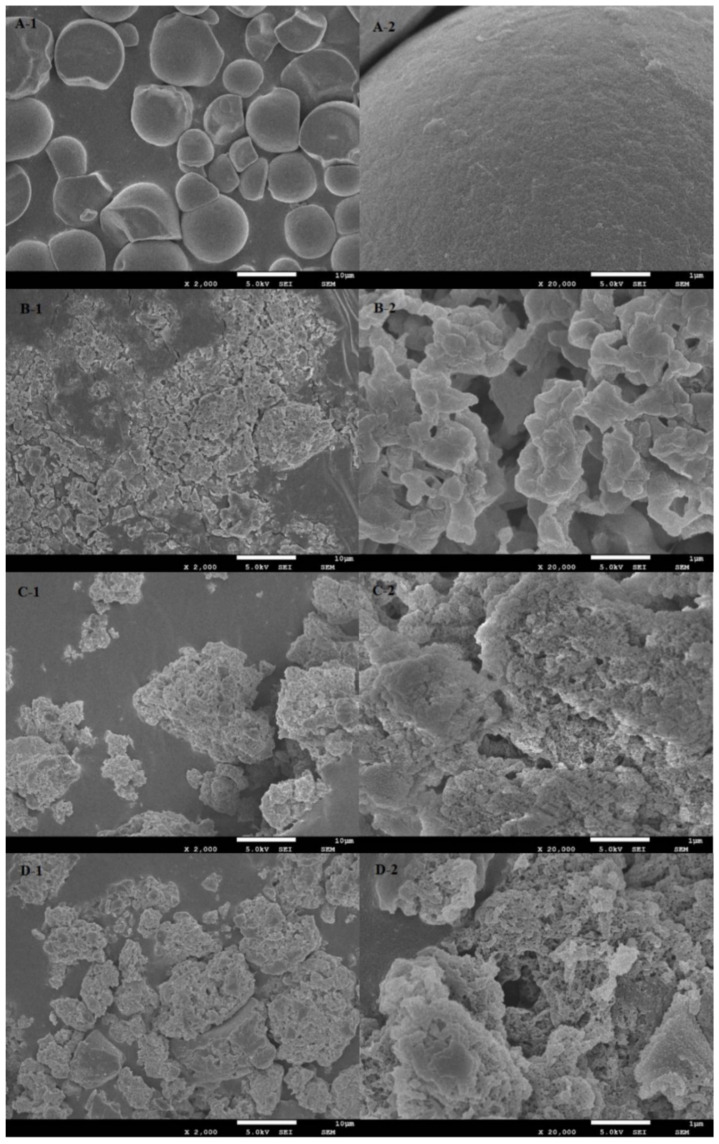
SEM photos of (**A**) cassava starch; (**B**) CA; (**C**) OSA-CA; (**D**) OSA-CA/Q, 1—×2000; 2—×20,000.

**Figure 2 molecules-26-06884-f002:**
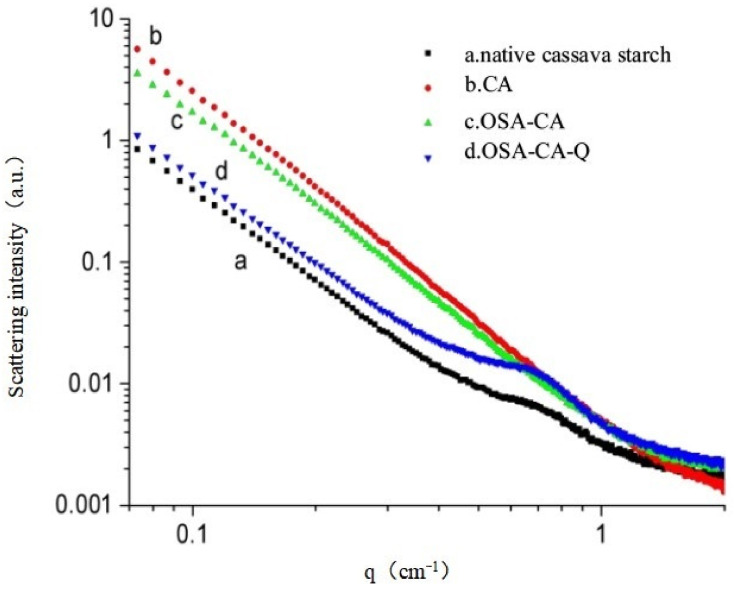
X-ray scattering curve. (**a**) native cassava starch; (**b**) CA; (**c**) OSA-CA; (**d**) OSA-CA/Q.

**Figure 3 molecules-26-06884-f003:**
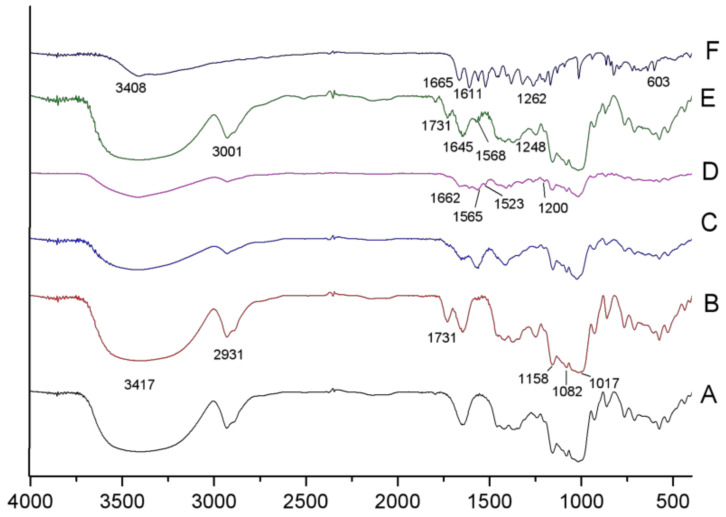
FT-IR spctra of (**A**) native cassava starch; (**B**) CA; (**C**) OSA-CA; (**D**) the mixture OSA-CA and Q; (**E**) OSA-CA/Q; (**F**) quercetin.

**Figure 4 molecules-26-06884-f004:**
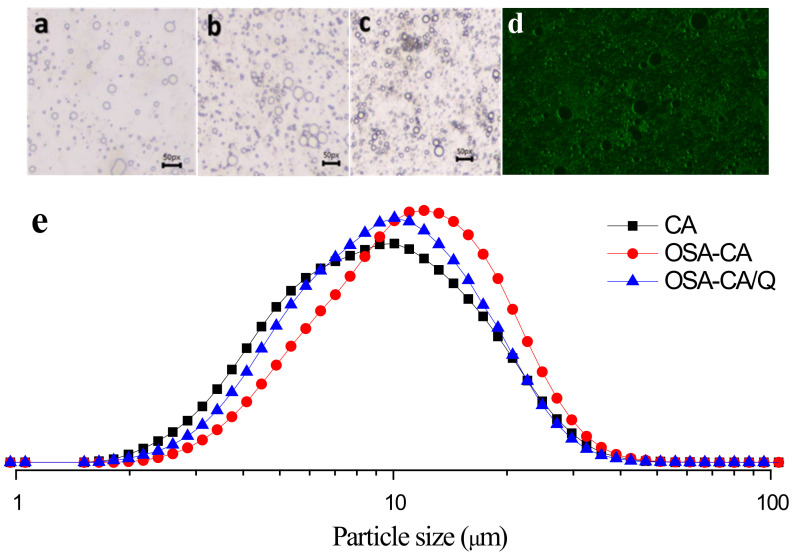
Micromorphology of emulsions. Scale bar = 50 μm. (**a**) PE; (**b**) SE; (**c**) TE and (**d**) fluorescence microscopy of emulsion and (**e**) effects of different emulsifiers on droplet sizes.

**Figure 5 molecules-26-06884-f005:**
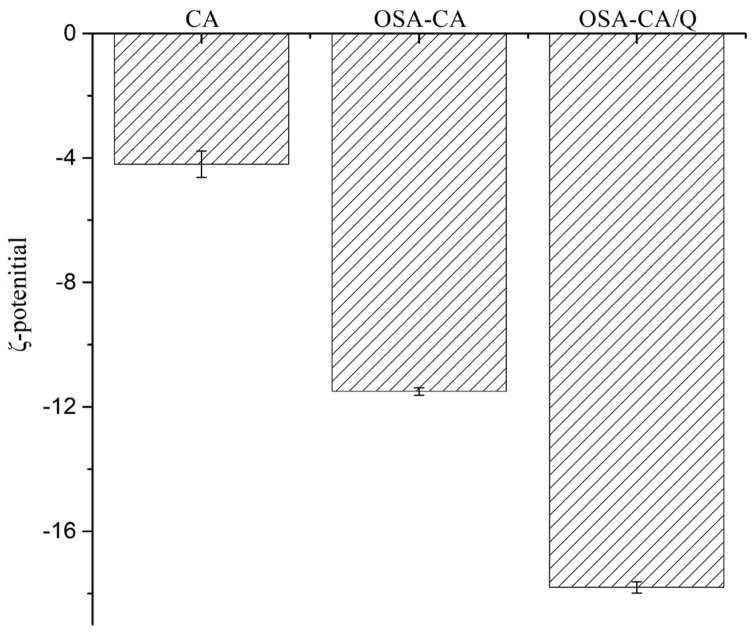
Zeta potential of different emulsions droplets.

**Figure 6 molecules-26-06884-f006:**
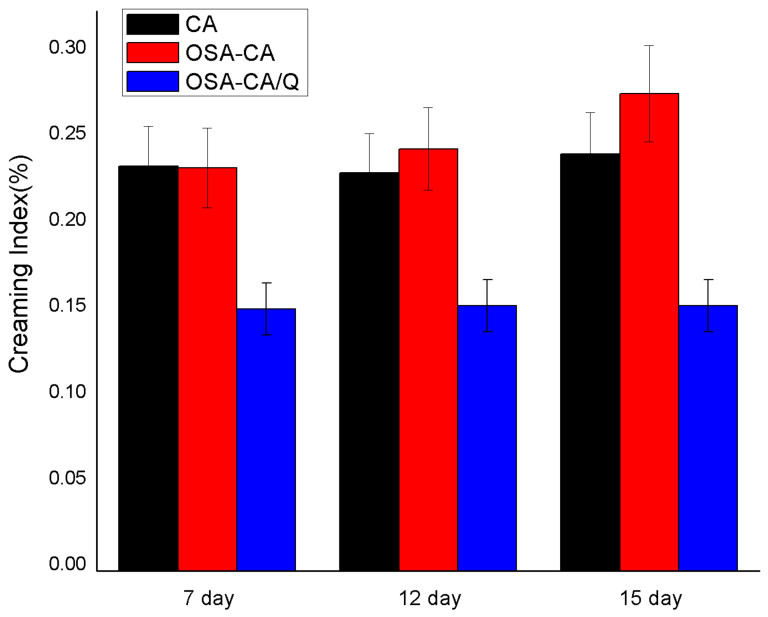
Evolution of the percentage of the creaming index (CI%).

**Table 1 molecules-26-06884-t001:** POV values of emulsions with different emulsifiers.

Emulsifier	POVs		
	5 d	7 d	15 d
CA	8.9	14.0	12.4
OSA-CA	5.4	10.0	8.9
OSA-CA/Q	5.2	5.0	5.4

## Data Availability

The data presented in this study are available on request from the corresponding author.

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
