# Peer review of "A Novel Functional Emulsifier Prepared with Modified Cassava Amylose with Octenyl Succinic Anhydride and Quercetin: Preparation and Application in the Pickering Emulsion"

_molecules, 2021, doi:10.3390/molecules26226884_

Round 1

Reviewer 1 Report

The article is well written and includes relevant and interesting scientific information. However, I think that the characterization of the emulsions could have been done much better. Not only do I mean other techniques, but to get more out of those used. In any case, I consider that with some improvements and answering some questions, the study is likely to be published in Molecules.

  • In my opinion there are too many abbreviations in the abstract.
  • Line 32. "... and antioxidants must be added" - Perhaps the authors should clarify whether the addition of an antioxidant is applicable to all types of emulsions or only to those with certain ingredients susceptible to this type of destabilization. In fact, usually a thickener is added to increase stability, not an antioxidant
  • Line 47. "...whipped cream [XXX]" - I think at least one reference should be added.
  • Lines 54-57. More current references to Pickering emulsions and the use of fumed silica or other compounds (eg zein) could be added.
  • Line 115. Please replace "x" by "·" in the equation
  • Line 134. Use wt% or (w / w) throughout the text, not both.
  • Section 2.5. Please indicate why this oil was used. What mass per batch was prepared? Also, why have you used these processing variables? Was it optimized in any way? It seems too long and at those homogenizatio rates recoalescence could occur.
  • Section 2.7. Do not use the abbreviation of SAXS in the title of the section
  • Section 2.11. I consider that if drop sizes have been determined (as the authors themselves indicate just below), the word "particle" should not be put in the section title.
  • Figure 4. It would be appropriate to reduce the x-axis from 1 to 100 micrometers.
  • Section 3.4. Analysis of droplet sizes is scarce. Not even mean diameter or polydispersity values ​​are indicated.
  • Section 3.7. Droplet sizes and polydispersity, as well as rheological properties, have a crucial influence on cremation rate. I suppose it is late to carry out rheological measurements, but it should at least be commented and references should be indicated.
  • Figure 6. No data for times shorter than seven days?

Author Response

Dear reviewer,

On behalf of my co-authors, we appreciate you very much for your positive and constructive comments and suggestions on our manuscript entitled “A novel functional emulsifier prepared with modified cassava amylose with octenyl succinic anhydride and quercetin: preparation and application in the Pickering emulsion”. We have studied your comments carefully and have made revision point by point which marked in red in the paper. 

Response to Reviewer 1 Comments:

Reviewer 1: The article is well written and includes relevant and interesting scientific information. However, I think that the characterization of the emulsions could have been done much better. Not only do I mean other techniques, but to get more out of those used. In any case, I consider that with some improvements and answering some questions, the study is likely to be published in Molecules.

In my opinion there are too many abbreviations in the abstract.

Response: The amount of abbreviations has been reduced according to your suggestion.

Line 32. "... and antioxidants must be added" - Perhaps the authors should clarify whether the addition of an antioxidant is applicable to all types of emulsions or only to those with certain ingredients susceptible to this type of destabilization. In fact, usually a thickener is added to increase stability, not an antioxidant

Response:Thank you for your excellent suggestion. The application of emulsifiers and antioxidants in emulsions has been discussed separately. The revised text is displayed in red font in the manuscript.

Line 47. "...whipped cream [XXX]" - I think at least one reference should be added.

Response:Thank you for your careful review. The related reference has been added to the manuscript and displayed in red font.

Lines 54-57. More current references to Pickering emulsions and the use of fumed silica or other compounds (eg zein) could be added.

Response:According to your suggestion, two references were added to the manuscript and displayed in red font.

Line 115. Please replace "x" by "·" in the equation

Response:Thank you for your reminder. This error has been corrected and is displayed in red font in the manuscript.

Line 134. Use wt% or (w / w) throughout the text, not both.

Response:Thanks for your information and for being so helpful. This error has been corrected and is displayed in red font in the manuscript.

Section 2.5. Please indicate why this oil was used. What mass per batch was prepared? Also, why have you used these processing variables? Was it optimized in any way? It seems too long and at those homogenizatio rates recoalescence could occur.

Response:Dear reviewer, thanks for your careful review. Camellia seed oil was chosen as the oil phase because camellia seed oil is rich in unsaturated fatty acids, which is particularly susceptible to oxidation. Therefore, camellia seed oil is particularly sensitive to the lipid oxidation index (POV value) that we measured. The mass of each batch is 100 g. Yes, the content of the oil phase has been optimized to achieve a more stable emulsion. Longer processing times are required to achieve smaller droplet sizes and more uniform emulsions.

Section 2.7. Do not use the abbreviation of SAXS in the title of the section

Response:Thanks for your reminder and it is very useful. This error has been corrected and is displayed in red font in the manuscript.

Section 2.11. I consider that if drop sizes have been determined (as the authors themselves indicate just below), the word "particle" should not be put in the section title.

Response:Thanks for your suggestion. This error has been corrected and is displayed in red font in the manuscript.

Figure 4. It would be appropriate to reduce the x-axis from 1 to 100 micrometers.

Response:Thank you for your constructive comments which was very helpful. The x-axis of the figure has been modified according to your suggestions.

Section 3.4. Analysis of droplet sizes is scarce. Not even mean diameter or polydispersity values are indicated.

Response:Dear review, thanks for your constructive comments. This is our negligence. Additional discussions are supplemented in this manuscript and shown in red font (Line241-243, Section 3.4).

Section 3.7. Droplet sizes and polydispersity, as well as rheological properties, have a crucial influence on cremation rate. I suppose it is late to carry out rheological measurements, but it should at least be commented and references should be indicated.

Response:Dear review, thanks for your constructive comments. Additional discussions and references are supplemented in this manuscript and shown in red font (Line269-273, Section 3.7).

Figure 6. No data for times shorter than seven days?

Response:Dear reviewer, thanks for your careful review. The storage period of more than 7 days is to obtain more obvious stratification, which is beneficial to obtain different CI indexes. This is more conducive to distinguish the differences between different samples. If the storage time is too short, the difference between the samples is not obvious.

Thanks again for your comments on our manuscript. Looking forward to hearing from you.

Yours sincerely,

Haiming Chen, PhD

Reviewer 2 Report

This is a well-written manuscript. I congratulate the authors for their in-depth study. I have no hesitation to accept the manuscript in its current form.

However, if possible, the authors may conduct the following studies:

  1. Even though authors have performed SAXS, it would be good if the authors include XRD
  2. The surface tension of the emulsions may be determined
  3. The rheological properties may be included

Author Response

Reviewer 2

This is a well-written manuscript. I congratulate the authors for their in-depth study. I have no hesitation to accept the manuscript in its current form.

However, if possible, the authors may conduct the following studies:

Even though authors have performed SAXS, it would be good if the authors include XRD. The surface tension of the emulsions may be determined. The rheological properties may be included

Response: Dear reviewer, thanks for your careful review and excellent comments. XRD is an effective technique for characterizing the crystal structure of starch molecules and more in-depth research will be conducted in the future. Surface tension has a significant effect on the adsorption of emulsifier molecules on the oil-water interface and the droplet size of emulsions. According to your valuable suggestions, more studies on interfacial and rheological properties will be carried out in the future. Thank you again for your constructive comments.

Round 2

Reviewer 1 Report

Taking into account the answers to the questions as well as the revisions made, I consider that the manuscript can be published in its current version.